# Emergency First Responders and Professional Wellbeing: A Qualitative Systematic Review

**DOI:** 10.3390/ijerph192214649

**Published:** 2022-11-08

**Authors:** Malcolm P. Bevan, Sally J. Priest, Ruth C. Plume, Emma E. Wilson

**Affiliations:** 1Flood Hazard Research Centre, Middlesex University, London NW4 4BT, UK; 2Department of Natural Sciences, Middlesex University, London NW4 4BT, UK; 3Nottingham Centre for Public Health and Epidemiology, University of Nottingham, Nottingham NG7 2RD, UK; 4Nottingham Centre for Evidence Based Healthcare, University of Nottingham, Nottingham NG7 2RD, UK

**Keywords:** emergency first responders, rescue worker, professional wellbeing, organisational culture, leadership, team building, welfare

## Abstract

Emergency first responders (EFRs) such as police officers, firefighters, paramedics and logistics personnel often suffer high turnover due to work-related stress, high workloads, fatigue, and declining professional wellbeing. As attempts to counter this through resilience programmes tend to have limited success, there is a need for further research into how organisational policies could change to improve EFRs’ professional wellbeing. Aim: To identify the factors that may contribute to or affect EFRs’ professional wellbeing. Methods: A systematic literature review has been carried out. Three databases (Science Direct, ProQuest, and PubMed) were searched using keywords developed based on the PICo (population, interest, and context) framework. A total of 984 articles were extracted. These were then critically appraised for the quality of the evidence presented, leading to a total of five being ultimately included for review. Results: Thematic analysis revealed that although EFRs may be exposed daily to traumatic events, factors that contribute to a decline in professional wellbeing emerge from within the organisational environment, rather than from the event itself. Conclusion: The study concludes that organisational and team relations factors significantly impact EFRs ability to cope with stress. As such, organisational policy should evolve to emphasise team relations over resilience programmes.

## 1. Introduction

Emergency First Responders (EFRs) are professionals that provide support and assistance in critical situations, including natural disasters, medical emergencies, accidents, and rescue situations. These professional categories include paramedics, police officers, firefighters, disaster recovery workers, and support personnel such as dispatch workers and coordinators. Considering their professional roles, EFRs are often exposed to critical situations which pose a danger to their physical and mental health. These events can have a significant impact on their professional wellbeing and subsequently, on the extent to which these professionals can carry out their roles.

Previous studies [1,2,3,4,5,6] indicate that as the need for EFRs is increasing, so do turnover rates and underperformance associated with burnout. Apart from the intense working conditions, these studies also link poor organisational leadership and negative self-regulatory strategies with professional turnover in these fields. Consequently, the specifics of EFR work environments create by default situations that put a significant strain on EFRs. Coupled with increased turnover rates, pressures on remaining personnel are further increased leading to reduced professional wellbeing. Noting these aspects, EFRs can be regarded as a high-risk category for poor professional wellbeing.

For the scope of this study, professional wellbeing is defined [7,8,9,10] as an aspect of the professional life comprised of social, psychological, physical and economic wellbeing. 

To address these issues, resilience training programmes have been proposed [6]. Resilience is briefly described as the capacity to overcome difficulties and adapt successfully under stressful circumstances [11]. However, as noted by various authors [7,8,9,10], professional wellbeing is comprised of several domains, including physical wellbeing, economic wellbeing, social wellbeing, and psychological wellbeing. As resilience programmes do not address these all these domains, these programmes are also not very successful. Hence, resilience programmes may be limited in their scope, as this approach may fail to address domains outside psychological wellbeing. As shown by previous interventional studies [11,12], resilience training is only moderately effective in reducing stress and improving wellbeing. 

As a result of the limited effectiveness of resilience programmes for EFRs, new developments must be made in research and practice to help improve the professional wellbeing of EFRs and reduce turnover rates. 

Noting these aspects, the aim of the current review of evidence is to assess the literature base and systematically identify and analyse qualitative data on factors that may contribute to EFR wellbeing and factors that affect professional wellbeing across all domains. The novelty of this study is focused on identifying and categorising factors that affect EFRs’ professional wellbeing, to enable the creation of future research directions for developing comprehensive interventions to support the wellbeing of EFRs. 

The study is structured in five main parts. The introduction presented the background to this study, the current research gap left by the intense focus of the literature on resilience programmes and the aim of the current review. The following section will present the methods used to extract literature for this review. Section 3 will present the results, followed by a discussion of these results in Section 4, and finally, concluding remarks in Section 5. 

## 2. Materials and Methods

Systematic reviews are qualitative methods used to identify and extract literature on a topic of interest, using a step-by-step process. The aim of this approach is to generate new meaning and knowledge from existing evidence and inform new research directions or/and highlight research gaps in current literature [13,14].

To set a clear research question for the present systematic review, the PICo (population, interest, and context) framework was used [13]. The framework was concomitantly applied to develop the inclusion and exclusion criteria for the articles identified, as well as the keywords used to identify this research [14]. The PICo framework is listed in Table 1 below. 

As observed from the above table the search strategy aimed to extract studies with participants across all emergency response domains. This included police officers, fire-fighters, paramedics, dispatch staff, search and rescue staff, and ambulance staff. The interest factors considered included any type of emergency or crisis situation to which EFR (emergency first responders) are exposed to. Finally, the context was considered within the UK, Europe, USA, Canada, and Australia. Considering the complexity of professional wellbeing domains, only qualitative or mixed methods research was selected for inclusion. 

Based on the PICo framework devised for this study, the main research question is: How does exposure to emergency and crisis situations affect first responders’ wellbeing?

### 2.1. Keyword Development

In line with the PICo framework, the main keywords used in the search process were: emergency responders’ wellbeing; first responders wellbeing; ambulance first responders wellbeing; police first responders wellbeing; fire-fighters first responders wellbeing; welfare. Boolean logic operators were used for combining the keywords: Emergency Responders OR first responders OR ambulance responders OR police responders OR fire-fighters responders AND (wellbeing OR welfare). 

### 2.2. Database Selection and Filters

Databases selected for the review had to reflect the specifics of the subject investigated [15,16]. In February 2022, an initial search process was carried out through the NUsearch library University of Nottingham, which includes Google Scholar, Web of Science and Scopus. No relevant data was extracted from this search process. Data from the Cochrane library was not used in this search process as only primary studies were sought to be extracted. Relevant adjacent data has been presented in the introduction section. 

As a result, Science Direct, ProQuest, and PubMed were selected as primary databases and used in one simultaneous search process. To limit the search results to the most relevant data, several filters were used during the search process [17]. Firstly, a time filter was applied to retrieve only data from the past 20 years (2002–2022). This timeframe was selected because more recent research was scarce and thus the time-filter was expanded. Secondly, only peer-reviewed journals and primary data were included.

### 2.3. Inclusion/Exclusion Criteria 

Inclusion and exclusion criteria were developed in line with the PICo framework [17]. These criteria are listed in Table 2 below.

A focus on qualitative articles was set for this review as the current investigation sought to identify aspects that affect professional wellbeing from a subjective perspective as described and experienced by EFRs. 

The application of the inclusion and exclusion criteria was carried out by two reviewers. Disagreements were resolved by discussions and reversion to the initial aim of this study. 

### 2.4. Search Results 

Results obtained following the search strategy described above are displayed in Table 3 below. The inclusion and exclusion criteria developed were applied to the articles identified through the aforementioned search process. 

A total number of 984 results were generated from which 263 were removed as duplicates. The remaining 721 articles were subjected to the inclusion and exclusion criteria based on title reads and abstract reads. A total of 657 articles were excluded. The remaining 64 articles were assessed for eligibility against the inclusion and exclusion criteria by full reading. A total of 59 articles were eliminated in this process either due to context mismatch, sample mismatch, or subject of interest mismatch. A total of 19 articles were identified, 14 were eliminated because of the use of quantitative methodologies. A total of five studies were thus included for review, from which three used a qualitative methodology [18,19,20] and two used a mixed-methods design [21,22]. The PRISMA Flow chart for study selection [23] is listed in Figure 1 below.

### 2.5. Data Analysis

Data synthesis was informed by the Cochrane manuals for systematic reviews [14,16,17], as well as by others [24,25]. The table summary is presented in Appendix A, Table A1. 

To critically appraise the selected studies JBI [25] tools were used for qualitative studies, including ConQual quality criteria. To assess the methodological quality of mixed methods studies, the tool developed by Hong et al. [26] Mixed Methods Appraisal Tool (MMAT) is used. GRADE Quality of Evidence developed by BMJ was applied for mixed-methods studies. The critical appraisal of the articles included is presented in Appendix B, Table A2, Table A3, Table A4 and Table A5. 

To further analyse the data extracted from the articles selected for review, thematic analysis [27] was used. Similar codes (words) were identified from the selected studies and grouped together to form subthemes. Subthemes were further condensed to obtain the main themes. This approach was selected in lieu of content analysis as thematic analysis allows for a more nuanced assessment of text data, including themes’ inter-relations and dynamics. 

### 2.6. Ethical Concerns 

This research does not include any human participants as the investigation is focused on a literature study. However, ethics pertaining to academic conduit still apply [28]. Consequently, no studies were eliminated based on the author’s country of origin or other discriminatory notions as pertaining to the Equality Act 2010. Data was extracted and analysed with the intention to improve current practice in EFR and professional wellbeing while enhancing professional stress management. A secondary peer-review process for the articles selected was conducted, thus minimising the risk of bias.

## 3. Results

### 3.1. Quality of Evidence 

Following the critical appraisal, all studies included for review achieved high scores (Appendix B). This indicates that the quality of the evidence is strong, as supported by high credibility and trustworthiness criteria demonstrated through the use of adequate methods and bias reduction strategies. Studies extracted were carried out in Australia, with only one study [18] carried out in the UK, and one [19] in the US. 

Two of the qualitative studies Coxon et al., Dropkin et al. [18,19] used semi-structured interviews and focus groups to collect data. The text data was analysed by qualitative thematic analysis whereby peer-reviewed methods of extracted codes and themes were applied. One study [19] used narrative interviews and quantitative thematic analysis via NVivo. The framework developed to analyse the data via frequency analysis in NVivo was subjected to peer verifications and adjustments prior to the application of the final framework for data analysis. 

Mixed-methods research Paterson et al., Pyper and Paterson [21,22] used a combination of semi-structured interviews and surveys. Eriksen [20] collected data via a survey with one open-end question. The text was analysed via qualitative thematic analysis with prior peer coding verification and agreements. Descriptive statistics were used to report on the quantitative data collected via the survey. Pyper and Paterson [22] similarly collected data by using previously validated questionnaires to assess for fatigue, anxiety, and depression, with an additional 10 open-end questions. Descriptive statistics were used to report on quantitative data. Manual deductive content analysis was used to analyse qualitative data. In both studies, qualitative and quantitative data were used adequately to provide a valid answer to the research question. 

As emerging from the data extracted, there is a limited in-depth evidence-base for factors that affect professional wellbeing in EFRs. Although the existing evidence was rated as high via MMAT and GRADE quality appraisal frameworks, it is unlikely that this evidence is sufficient to produce complex organisational and professional interventions which may help in reducing the negative effects of the EFR profession on wellbeing.

### 3.2. Thematic Analysis 

Three main themes were extracted from the studies included for review; “Organisational Challenges”; “Professional Challenges” and “Community”. A thematic map (Figure 2 below) was developed to illustrate the relationships established between the themes identified, their subthemes, and codes. Based on the analysis of the main themes extracted, it was observed that participants in all studies rarely mentioned being affected on a psychological level by attending emergency services work. Instead, most commonly cited issues related to organisational contexts, as well as professional challenges. Moreover, these professional challenges seemed to further be amplified by organisational issues that produced various factors which affected professional wellbeing across physical, psychological and economic domains. 

As illustrated above, organisational challenges amplified professional challenges, while community and social support perceptions reduced psychological distress in certain conditions. Professional satisfaction buffered out the effects of stress and emerged from leadership and team member relations, as well as from being able to provide help to people in emergency situations or disaster recovery. However, when team relations were negative, characterised as perceptions of lack of value, recognition, and respect among team members, professional satisfaction declined. These notions are discussed in depth in the following sections. 

#### 3.2.1. Organisational Challenges 

As emerging from the analysis of qualitative studies Coxon et al., Dropkin et al. [18,19], organisational challenges seem to arise from interactions with co-workers, leadership, and management, as well as from available resources, including available equipment, training, and pay. Organisational challenges and professional challenges have been suggested to be addressed by the provision of adequate training, more work-related autonomy, as well as social support, and better leadership. 

##### Team Relationships

Team relations and relations with supervisors and leaders have been identified as contributors to work-related stress and anxieties. When relationships are poor within the work force, these aspects seem to hinder recovery and resilience, while adding to burnout and fatigue. As a result, social support and good working relations were noted as potential buffers to professional stressors. 

Perceptions of dispatch staff over main job stressors were identified Coxon et al. [18] as emerging from interactions with field team members, including lack of acknowledgment and respect, as well as lack of collaboration. This has been explained by participants as being caused by the fact that dispatch staff does not physically participate in emergency situations. Hence, other EFRs have a devalued perception of their contribution to the emergency service, and a poor understanding of their organisational roles and attributes. 


*“All of the participants mentioned feeling overlooked, misunderstood and marginalized. When specifically discussing the public perception, participants described feeling invisible.”*
[18]

These perceptions from other EFRs, as well as their effects on dispatch staff, seem to erode working relations, creating conflicts and contributing to job-related stress. Furthermore, although team-building exercises were carried out within the organisation, participants felt that these training sessions did not address what was really important and relevant for their context. Subsequently, some argued that these exercises amplified their feelings of being devalued, as they felt that their time was ill-spent. This indicates that when organisations aim to develop team-building programmes, these programmes need to address the specificities of the EFR profession. 

Issues in co-worker relations, and in particular, relations established with leadership representatives have also been reported elsewhere, Dropkin et al. [19] following interviews with a sample of emergency medical service workers (EMS) and their supervisors. In this case, lack of trust between leaders and EMSs, as well as lack of consideration from leadership in relation to the safety needs of EMSs were noted as main organisational challenges. Moreover, organisational policies for employing new EMSs contributed to an increase in job-related stress, as an initial screening process for physical fitness was lacking. This was a main concern for EMSs as they were also unable to choose a shift partner and were thus left with carrying out heavy lifting during patient transport on their own. 

##### Organisational Resources and Salaries 

From the qualitative data assessed Coxon et al., Dropkin et al. [18,19], insufficient pay and limited organisational resources have also been reported as contributors to stress. For example, as noted by one dispatch participant [18], the lack of availability of ambulances while taking an emergency call creates significant pressure and stress for the dispatch workers. 


*“In dispatch, the main stressors are not having the resources, in terms of ambulances for emergencies. (Jane, female, 13½ years in role).”*
[18,19]

Similarly, other similar investigations Dropkin et al. [19] found that limited resources available for staff training and equipment are acknowledged by supervisors and EFR employees alike, yet solutions are lacking. EMS further acknowledged that ergonomic training is relevant for preventing injuries, while supervisors note that resources for this type of training are scarce. 


*“Supervisors also noted that workers needed more training on waiting for help (either from another EMS team or a supervisor) and not rushing on the last call in order to go home. However, they noted there was no scheduled time for additional training on appropriate and safe work practices.”*
[18,19]

Participants in qualitative studies [18,19] reported underpay as a main issue related to job dissatisfaction. Moreover, underpay forced many EMSs to take on second jobs. In return, this impacted on their ability to fully rest after shifts, especially following night shifts. 


*“Initially, when asking Darren what his routine was after finishing work, he confessed to not really having a routine. As Darren explained, he often returns to work not feeling refreshed, and he now does not experience the same enjoyment in his role as he used to.”*
[18,19]


*“Low pay, work shifts, and second jobs were related according to workers. Low wages led to second jobs, but most had no choice because of their salaries.”*
[18,19]

##### Work Schedules and Shifts 

Several studies [18,19,20,21] identified shifts and work schedules as main challenges for EFRs. In this case, night shifts, workload, poor sleep-wake patterns, understaffing, as well as working second jobs were found to decrease job satisfaction and increase the likelihood of burnout and fatigue. These issues were significantly amplified for participants who also had family responsibilities. As noted by participants in these studies, poor dietary habits were common, as limited breaks and busy working schedules do not allow for proper meals during shifts. Additionally, working second jobs impeded EFRs to rest properly between shifts, with family life further adding to fatigue and exhaustion. As reported by this research [18], while EFRs may take shift breaks, especially after a stressful situation, when they get back on the clock, they often find themselves in a completely different crisis situation. Quickly adapting to the new scene has been reported as an additional stressor. With heretic shift schedules, second jobs, and the stress of work, many participants found it hard to relax after their shifts and have an adequate work–life balance. Concomitantly, others noted that family stress further added to their feelings of fatigue. 

#### 3.2.2. Professional Challenges 

Specific professional challenges related to emergency services were reported by all studies included for review [18,19,20,21,22]. These challenges included the need for specific training and equipment, environmental risks, working with heavy loads, as well as balancing professional and social lives. Considering that the samples in the studies included for review consisted in majority of emergency medical service workers, it can be argued that the identified professional challenges may apply only to this profession. Hence, additional challenges may be present in other EFR professions. 

Environment risks identified derive from weather conditions, but also from the specifics of the emergency where EFRs must provide assistance. For example, emergency response workers are often exposed to bushfires while carrying out their duties as part of rescue and rebuild missions Eriksen [20]. Weather conditions were reported as physical exposure factors that can result in physical injuries Dropkin et al. [19]. The availability of protective equipment seemed to mitigate these risks Coxon et al. [18]. 

Rural ambulance staff perceived that they are exposed to harm, especially when lacking protective equipment against fires Pyper and Paterson [21]. 


*“These responses included being located in a ‘rural area’, being exposed to ‘hot weather’ and being ‘not adequately prepared for heat waves/bushfire crises.”*
[21]

Other physical risks identified include musculoskeletal injuries that are attributed to carrying heavy loads during patient transportation or while carrying heavy equipment [19]. 


*“Once I have an injury, just one wrong move with a patient or when carrying equipment will flare me up. I can lose three months of work.” Most supervisors reported that weight and number of pieces of equipment that EMS staff and their partners carried and handled, combined with icy, wet weather were ‘headaches’.”*
[19]

Some amplifying effects were observed, as being in physical pain while on the job further amplifies psychological stress in this group [19]. Consequently, these aspects denote that EFRs may be exposed to further decline in psychological wellbeing when sustaining a physical injury. 

At the other end of the spectrum, working in emergency services in rural areas may be particularly challenging, as in some cases EFRs may need to work alone, while at the same time depend on the response time of metropolitan services [22]. 

#### 3.2.3. Community

In two of the studies identified Eriksen, Pyper and Pateron [20,22], the community served by EFRs acted both as a motivator and social support factor, but also as an additional stressor. In supporting populations through disaster recovery, workers in disaster recovery were noted to develop social connections with people from the affected community, which further contributed to the development of a sense of collectiveness and faith in humanity [20]. 


*“Faith driven by group-identity connects to collectively created mental, spiritual and physical spaces and practices where it is ‘safe’ to confide, reflect, debate, grow and heal through interaction with people who share a common purpose, belief or relational support.”*
[20]

These effects were noted to buffer out stress from team tensions, as well as stress emerging from risks of exposure. Concomitantly, within these social connexions, disaster recovery workers felt a sense of purpose that allowed them to withstand physical and mental stress. Despite these observations, rigid hierarchical structures, power dynamics, and imbalances can create additional psychological stressors for recovery workers [20]. 

These findings are not singular, as dispatch workers reported feeling personal and professional satisfaction when they were able to provide life-saving support to people but felt devalued by other EFRs [18]. A sense of purpose and professional pride was also reported by participants, which further contributed to decreasing work-related stress. 

However, results from research using small regional EFR samples [22] indicates that working in small communities can be both a factor that improves professional satisfaction, as well as an additional work-related stressor. 


*“While several respondents stated that treating personally known patients is a source of significant stress, it was also identified that this can be rewarding. This is largely due to a ‘higher expectation to perform well’ and because ‘many (patients) are known friends or family’. However, it was also reported that ‘treating personally known patients who are not in a serious condition is easier and quite rewarding’.”*
[21]

In this case, interviews with regional ambulance staff workers showed that by being able to offer support to people that they know personally, participants felt a sense of professional and personal satisfaction. Nonetheless, when treating seriously ill patients that are known personally by ambulance staff, stress is significantly increased. 

## 4. Discussion

The nature of the literature available in this systematic review demonstrated that there is a stress issue in EFR which aligns with other quantitative studies [12,29] describing similar trends in police and firefighters, building a bigger body of evidence that this is a wider occupational issue in EFR roles. This systematic review aimed to systematically identify and analyse data on factors that may contribute to EFR wellbeing and factors that affect professional wellbeing across all domains. However, there is limited qualitative data in these groups. Furthermore, while themes may be transferable to different EFR groups, there is not sufficient evidence without further research with these groups to generalise. 

Despite the fact that this review identified 19 articles investigating EFR professional wellbeing, only five studies assessed factors that hinder or contribute to professional wellbeing via qualitative methods. Concomitantly, two of these studies Paterson et al., Pyper and Paterson [21,22] used mixed methods research. This indicates that while the quality of the evidence is high, there is also a significant lack of depth into understanding wellbeing in EFRs, as there is a general preference for quantitative studies that either measure burnout and/or fatigue via various instruments, or test interventions to address these issues. Due to this general literature trend, there is a limited understanding of EFRs perceptions of factors that affect their professional wellbeing, as well as of factors that may prove to be supportive in limiting negative wellbeing effects. 

Moderate effects of interventional studies [7,8,9,10,11] for improving wellbeing may thus be explained by a lack of understanding of how these factors emerge, interact and actually impact on EFR wellbeing. Concomitantly, as these studies focus on building resilience within the EFR workforce, the proposed solutions fail to acknowledge the contribution of the organisational environment to professional wellbeing. Consequently, this aspect remains unaddressed by current studies from the perspective of improving overall wellbeing, rather than focusing only on resilience. Similarly, it is questionable that approaches based on resilience alone can be successful on the long term when organisational aspects are not addressed, as these seem to play an important role in building EFR wellbeing at work. These aspects can explain why, despite current interventions to improve professional wellbeing, current rates of turnover still remain high. 

As observed from the thematic analysis, all domains of professional wellbeing [10] are affected to a certain degree by working as an emergency first responder. Nonetheless, contrary to the belief that psychological distress from being exposed to traumatic events [7] is a main contributor to psychological wellbeing decline, organisational factors also seem to play a significant role. 

In this sense, fatigue and burnout seem to be directly caused by shift scheduling and by organisational policies relating to breaks and rest times that do not allow EFRs to fully unwind after their shifts Coxon et al. [18]. Fatigue and burnout seem to be amplified by the need to work a second job, thus demonstrating a direct link between economic wellbeing and psychological wellbeing in the studied population. Concomitantly, as reported by others [29,30], fatigue may result in decreased attention to safety behaviours, which in return may result in physical injury while on the job. This indicates that adequate rest and pay are essential to job performance for EFRs as these elements address aspects of economic, physical, and psychological wellbeing. 

Organisational resources, in terms of staffing, training, and equipment, further increase the risk of physical injury Dropkin et al. [19]. Hence, a connection with physical wellbeing emerges, as exhaustion not only decreases physiological wellbeing but also exposes EFRs to potential physical harm, which is further amplified when there are insufficient resources to ensure protection. 

Noting that physical injuries can increase physiological distress for EFRs, a connection between physical and psychological wellbeing is observed. Moreover, others [30] report that increased psychological stress, either attributed to the specifics of EFR work, or to organisational factors, can trigger negative coping strategies for EFRs, such as substance abuse, thus adding to physical risks and declining physical and psychological wellbeing. 

One factor that has been repeatedly reported as having a buffer effect for stressors that decrease professional wellbeing is social support. However, when social support is replaced by internal team conflicts and lack of trust between supervisors and EFRs, social contexts create unnecessary work-related stress Coxon et al., Dropkin et al. [18,19]. A poor work–life balance and family responsibilities also act as amplifiers to psychological distress. In small communities, a social connection between EFR team members and community members increases personal and professional satisfaction, while minimising physiological distress Paterson et al., Pyper and Paterson [21,22], despite potential psychological pressure added from social relationships between EFRs and people. 

Considering the data reported by the studies included for review, factors that affect professional wellbeing can be grouped as organisational factors and professional factors. In this sense, poor organisational policies in shift scheduling and breaks, limited training, and limited resources can decrease professional wellbeing. Similarly, poor organisational cultures that do not emphasise mutual respect and collaboration can erode work satisfaction and contribute to burnout. Professional risks arise naturally from the work-related duties of EFR, including weather conditions, carrying heavy loads, and being exposed to physical harm. 

Herein, available protective equipment may buffer these risks, as well as adequate rest times that enable EFRs to act in line with safety measures. Other protective factors as emerging from these data are social support, and professional and personal satisfaction with the work carried out.

### Limitations 

Findings in this study need to be considered with some limitations. Firstly, the available data may be transferable only to medical emergency services, as only one study by Eriksen [20] included in the review had a mixed sample of disaster recovery workers and EFRs. All other studies used ambulance dispatch staff, paramedics, or EMS workers. Generalisability to the UK may also be limited as only one study [18] was carried out in the UK. 

Despite these limitations, it can be hypothesised that environment and physical risks may be significantly amplified for other EFRs who are consistently exposed to dangers, such as fire department workers or first responders in case of natural disasters. Secondly, because of the limited number of studies included for review, other themes that can be related to professional wellbeing may have been missed. 

Additional research may have been available through reference list scanning of the initial 19 articles identified (from which only five were included in this study based on the qualitative criteria). Additional studies focusing on this topic should thus consider expanding the search process through reference list scans. 

Finally, this study may be limited by the choice of keywords used in the search process. Emergency services have various categories of professionals that may have been used as synonym concepts in the search process to produce more results. 

## 5. Conclusions

By the nature of their profession, EFRs are exposed to traumatic events as part of their job roles to varying degrees. However, while turnover rates are increasing [1,2,3], the data emerging from this systematic review suggests that job-specific demands are not the main cause of this trend. Conclusively, being exposed to traumatic events as a result of EFR job roles is not a main contributor to declining professional wellbeing, and subsequent turnover. Instead, organisational stressors seem to play an important part in affecting wellbeing, by impacting various aspects of economic, social, physical, and psychological wellbeing domains. Moreover, because the majority of factors that result in professional wellbeing decline emerge from within organisations, this also implies that these factors can be controlled. 

Firstly, organisations are responsible for fostering a culture of cooperation and collaboration, as well as mutual respect and trust between employees. In EFR organisations, when these aspects are lacking, significant psychological distress is experienced by employees. Lack of trust between supervisors and EFRs, as well as a poor understanding of professional roles among EFRs seem to increase work-related stress and decrease job satisfaction. Organisational resources have also been observed to impact economic wellbeing, as well as physical wellbeing, especially when pay perception and safety equipment availability are low. 

Nonetheless, because organisational factors seem to contribute significantly to professional wellbeing decline, it can be assumed that restoring professional wellbeing also falls with organisational responsibilities and abilities. Consequently, building better work cultures, strengthening team collaborations, and developing strong leadership frameworks could potentially improve professional wellbeing. Strategies aimed at building better EFR team relations, and improving organisational cultures, may thus act as proxies to building better and more supportive social environments. Better social relations at work could thus contribute to improving professional satisfaction by aiding recognition and value among team members. Addressing these aspects rather than focusing on resilience alone may thus be more effective in reducing turnover rates in these professions and improving wellbeing among the work force. 

However, strategies that could be developed to address these aspects lack supporting research and evidence. Future efforts should thus be focused on researching further the factors identified in this study as contributors and hinders of professional wellbeing, while additional research investigating and testing various organisational strategies that can lead to improved wellbeing are also necessary. 

## Figures and Tables

**Figure 1 ijerph-19-14649-f001:**
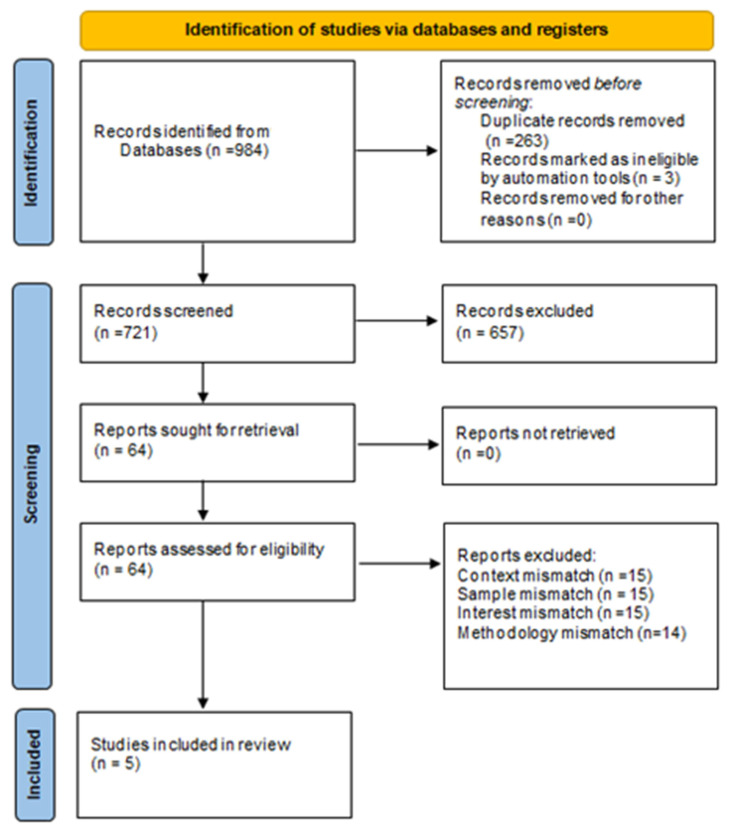
PRISMA Flowchart.

**Figure 2 ijerph-19-14649-f002:**
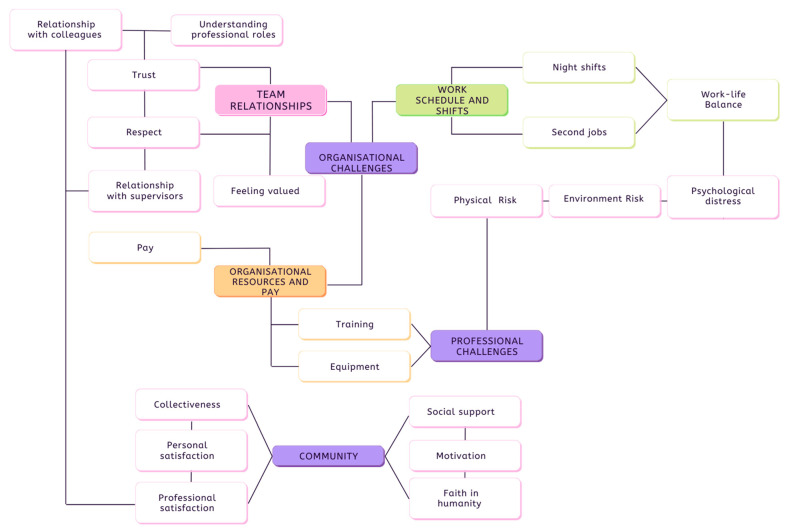
Thematic Map.

**Table 1 ijerph-19-14649-t001:** PICo Framework.

Population	Emergency responders across all domains (fire, police, emergency first responders)
Interest	Emergency and crisis situations, flooding, psychological, wellbeing
Context	Primary context: UK, United Kingdom, England, Wales, Scotland, Northern, and Ireland. Secondary Context: Europe, USA, Australia, and Canada.

**Table 2 ijerph-19-14649-t002:** Inclusion/Exclusion Criteria.

Inclusion	Exclusion
Study sample includes all categories of emergency response personnel: fire-fighters, ambulance staff, police staff, medical staff responding to crisis, and dispatcher for emergency situation calls.	Study sample is not specific to EFR. This may include students in training for becoming EFR, medical staff in local hospitals unrelated to community crisis situations, or dispatch for suicide help lines.
If other staff is included in the research, then data for EFR is separately reported and analysed.	Studies including volunteers as emergency first responders, or studies for which professional EFR samples have not been separately analysed.
Studies assessing contribution factors to wellbeing deterioration among EFR.	Studies assessing the prevalence of cautions caused by exposure to crisis situations in EFR.
Studies assessing, wellbeing as related to professional/organisational, emotional, physical and psychological wellbeing.	Studies focusing on wellbeing domains outside the context of emergency situations.
Studies assessing interventions to improve all or one aspect of wellbeing for EFR attending crisis situations.	Studies focusing on interventions to improve response times in EFR or procedural interventions.
Studies assessing the impact of crisis situations on EFRs wellbeing.	Studies assessing the impact of crisis situations on local populations.
Published within the past 20 years.	Published before 2002.
Published in English.	Published in a language other than English.
Empirical study/primary research.	Secondary research studies/commentary articles.
Qualitative and mixed-methods studies	Quantitative articles

**Table 3 ijerph-19-14649-t003:** Search Results.

	Databases		
Keywords	Science Direct	PubMed	ProQuest
Emergency responders’ wellbeing/welfare	205	4	7
Ambulance first responders’ wellbeing/welfare	249	1	5
Police first responders’ wellbeing/welfare	259	2	2
Fire-fighters first responders’ wellbeing/welfare	245	2	3
**Total: 984**			

## Data Availability

Not applicable.

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
