# Peer review of "Emergency First Responders and Professional Wellbeing: A Qualitative Systematic Review"

_ijerph, 2022, doi:10.3390/ijerph192214649_

Round 1

Reviewer 1 Report

Thank you for the opportunity for reading this paper. In my opinion, the paper requires improvements to be published. I would share some considerations for improving the quality of the article. Please see the comments below.

Abstract is to be built the following way: the purpose of the article, methods, results, conclusions, and recommendations/future directions and implications. Now the elements of novelty, the purpose of the article, future directions and academic and practical or political implications are not mentioned, also what is done by the authors.

Introduction:

-        the research gap, purpose and aims of the research are not clear.

-        Now the elements of novelty and purpose of the article are not clear, what is done by the authors? The "Introduction" section lists all relevant publications relevant to the study topic; it is noted what problems have already been discussed, and what remains unsolved, and the general area of the author's own study is determined. Thus, the place of own study in a specific field of knowledge is defined. This section should contain a justification of the need and relevance of the study

-        Literature review is too narrow and it is reflected also by a short list of the literature reviewed. A number of resources is not a criterion by itself but the research field is not revealed clearly and the research gap is not focused.

-        The structure of the publication is to be explained at the end of the Introduction.

Methodology:

-        The systematic literature review has certain requirements. It is useful to present how this method is used in similar research in terms of the research field, the scope of publications researched, etc.

-        Science Direct, ProQuest and PubMed are searched to collect the publications to research. There is a question of the inclusion of those databases and excluding databases such as Web of science and Scopus. A methodological explanation is required to understand and prove the choice.

-        Thematic analysis is used as a method to analyse the publications. The methodological explanation is required to prove the selection of this method against others such as content analysis or others that also can be used to analyse texts and contents.

Discussion“

-        the section needs to be expanded by debating the results presented in the publication with a wide range of other scholars analysed in the literature review section where the research gap is to be revealed.

Conclusions:

  • Although the authors present the added value of this paper, the authors might wish to consider explaining the practical and academic implications should be presented.
  • There is the possibility of future studies based on this analysis.
  • A future research agenda needs to be provided in the concluding remarks.

Author Response

Thank you for your constructive comments. Please see the attached word document for the response to reviewer 1.

Reviewer 2 Report

Thank you to the authors for sending in this paper.  I have minor comments: 

Methods:

Could you please give the exact years for each database searched?

Could you clarify whether a second author screened and participated during the data charting/extraction?  If not, then could you please have a second author check some of the studies included.  

Results:

Which countries were these five studies based in?  

Discussion:

Lack of generalizability - could you indicate this limitation in your Discussion section and expand on it.  There are five studies included, and as such, it is difficult to generalize to the rest of the populations based on this number.  Please also mention the need for further qualitative studies on this topic. 

Tables at the end of the manuscript:

Could you please change the page layout from portrait mode to landscape - this will lead to better formatting/viewing.

Author Response

Thank you for your constructive comments. Please see the attached word document for the response to reviewer 2.

Reviewer 3 Report

- Reviewer’s Commentary –

Thank you for submitting the manuscript for review.
The review evaluated qualitative studies on professional wellbeing and its influencing factors of Emergency First Responders. The result is that organizational factors are predominantly an influencing factor. Overall, there are significant weaknesses in the methodology and in the discussion. No information was provided on author contributions, which I find important especially in the context of a review. Especially the fact that there were at least two reviewers who evaluated und discussed the included und excluded studies. This information is missing, which weakens the paper considerably. While reading the paper, many thoughts occurred to me, which I would now like to present to you. Best succes for revision. It should be reviewed again afterwards.

General comments:

1) Please add the “authors contributions” after the conlusions and before funding. They were also listed in the template. This information is required.

2) Please adjust the citations in the text in accordance with the author guidelines: square brackets, not superscripted.

Abstract
3) Please add to the abstract: Methods, Results, and Conclusions.
4) Please also complete the professional groups of the EFRs in the abstract.
5) Lines 17, 20, and 25: Delete the large gaps in the text. Are those too many spaces?
6) First sentence: Please rephrase. Burnout, etc., is not the result of the additional need for emergency responders, but of the high workloads.

Keywords
7) For better finding in literature databanks, it is recommended to use other keywords than those in the title or abstract. My recommendations: emergency first responders → rescue workers, professionsal wellbeing → welfare

Introduction
8) Lines 39/40: see commentary No. 6.
9) Lines 45/46. Because it is the main topic of your publication, please explain what professional wellbeing and resilience is.

Material and methods
10) Write your review according to PRISMA-ScR Checklist (Tricco et al. doi: 10.7326/M18-0850). It leans on the author guidelines. Fill in the missing information.
11) Have you registered your scoping review (study protocol) somewhere?
12) Table 1: Explain the abbreviation EA, NRW, SEPA, NIEA...
13) Explain the reasons for your choice of included nations. This is a major limitation and restricts the message of your results.
14) Explain the choice of databases. Have you looked in the Cochrane Libaray for reviews that address the topic so you can find suitable literature?
15) Explain the time filter. Why the past 20 years?
16) When was the literature search conducted? Please specify exact date.
17) Which keyword should represent prehospital emergency medical services? I find this selection to be inadequate. The selection of keywords should be listed as a limitation. Paramedics or prehospital were better than ambulance.
18) Please add: who did the literature search. How many reviewers analyzed the sources? Two reviewers? And who did it? Who analyzed the publications? What was the procedure in case of disagreement? Who was the 3rd reviewer? When doing so, write 1st author, 3rd author, or use the initials of the names.
19) Was no hand search performed? So publications from websites or from the reference list? If not, add it as a limitation.
20) Lines 90/91: Should be transered in the table 2.

21) Table 2: Was there a minimum number of subjects in the studies examined? If applicable, please add it to Table 2. Also add the type of study: qualitative or quantitative.
22) It is unclear to me whether the search was performed separately in each case or whether all search terms (such as Lines 81-83) were performed in one search query. Please specify.
23) Lines 103-112: See commentary No. 14; please add the information.
24) Ethical concerns: Can be deleted here. It can be inserted under the annotations after conclusions.

Results:
25) Please add the occupational groups that were studied exactly in the publications.
26) Figure: Thematic Map of what? You can explain all the figures and tables in a little more detail. You have enough space.
27) Lines 195-199: Please add the reference.
28) Line 204: Delete the large gap in the text
Discussion
29) Lines 332: Please chance into scoping review.
30) 26) Lines 335-337: It is not interesting to mention the 19 studies here. You have identified 5.
31) Overall, I find it lacking a critical discussion of the results with existing literature. Since police and firefighters were hardly considered here, other studies could be discussed. Thus, transferability to these occupational groups could be suspected
if necessary.
32) Delete the large gaps in the whole text.

Limitations
33) Lines 394-396 and 401/402: See commentary No. 28.
34) The limitation should be completed: Selection of countries, time period(?), possibly only one reviewer, selected databases, lack of hand search of the literature...

Conclusions

35) Lines 405-406: “...frequently to traumatic events...” Frequently? Do you have a number for the individual occupational groups? I wouldn't generalize here. It varies greatly from one occupational group to another. However, the large number of ambulance calls is also because there are many unauthorized calls because of the convenience of the patients. That is not primarily traumatic. As an active emergency physician with over 3,000 calls, few are traumatic - thank God and carry on! Of these, also as chief emergency physician for Mass Casualty Incident (MCI).
36) You still need to discuss that the outcome of wellbeeing also depends on coping and personality.
37) You should support and discuss your findings with quantitative study samples. Appendix A:
38) Please present all tables in the appendix in a readable manner.
39) Write the tables in a uniform way. Why column 1 from Eriksen et al. and not before?
40) Keep the tables in key points and do not write a summary of the studies. It is unnecessary. And limit yourself to the key statements. For example, it is irrelevant whether the interview was grounded theory. It should only be presented occupational group of the EMR, the number, gender if applicable, the country structured or semi-structured interview. Then core question and key point results. Also write it one below the other and not in columns. This is hard to read.
41) Explain how you arrived at the degree? It is impossible to understand. Appendix B.
42) The titles of the studies do not have to be listed (e.g. Table B1, B2...). Write the first author (et al.) there.
43) What should the Y/U mean? Explain the figures. I think it is also useful to show the evaluation tools as well or to show what Q1, Q2, etc. mean.

Author Response

Thank you for your constructive comments. Please see the attached word document for the response to reviewer 3.

Round 2

Reviewer 3 Report

Thank you very much for the revision of the manuscript. I wish you all the best.